# Precision arbovirus serology with a pan-arbovirus peptidome

William R. Morgenlander [1,2], Wan Ni Chia [3], Beatriz Parra[4], Daniel R. Monaco[1,2], Izabela Ragan[5], Carlos A. Pardo [1,6], Richard Bowen[5], Diana Zhong [7], Douglas E. Norris [8], Ingo Ruczinski[9], Anna Durbin [10], Lin-Fa Wang [3], H. Benjamin Larman [1,2] ✉ & Matthew L. Robinson [7] ✉

Arthropod-borne viruses represent a crucial public health threat. Current arboviral serology assays are either labor intensive or incapable of distinguishing closely related viruses, and many zoonotic arboviruses that may transition to humans lack any serologic assays. In this study, we present a programmable phage display platform, ArboScan, that evaluates antibody binding to overlapping peptides that represent the proteomes of 691 human and zoonotic arboviruses. We confirm that ArboScan provides detailed antibody binding information from animal sera, human sera, and an arthropod blood meal. ArboScan identifies distinguishing features of antibody responses based on exposure history in a Colombian cohort of Zika patients. Finally, ArboScan details epitope level information that rapidly identifies candidate epitopes with potential protective significance. ArboScan thus represents a resource for characterizing human and animal arbovirus antibody responses at cohort scale.

Emerging infectious diseases repeatedly threaten global health. Many viruses with pandemic potential are transmitted by arthropod vectors. In recent decades, outbreaks of multiple such arboviruses have either expanded substantially, such as dengue[1], have emerged in new locations (e.g., Zika virus[2]), or have been newly discovered (e.g., Heartland virus[3]). Global health organizations have therefore emphasized the importance of surveillance for emerging arboviruses and development of countermeasures, such as vaccines[4], to mitigate arboviral infections.

Arbovirus surveillance is technically challenging for several key reasons. A genetically diverse pool of hundreds of known arboviruses limits the scope for nucleic acid tests that may simultaneously target only one or several specific arboviruses. Furthermore, many arbovirus infections only briefly produce viremia, such that sampling for nucleic acid of the virus must occur during a brief window, often at illness onset. Serologic assays may demonstrate prior exposure, but available assays suffer from cross-reactivity and/or are time- and cost-intensive to perform[5,6]. New approaches are needed for arbovirus serology that distinguish related viruses, enable higher sample throughput, and incorporate both human and zoonotic viruses.

In addition to enhancing surveillance capability, improved arboviral serological tools will deepen understanding of host immunity. In the context of flaviviruses, an individual who has been infected with an initial dengue serotype is at increased risk for severe dengue if infected by a different dengue serotype years later. Cross-reactive but non-

[1]Department of Pathology, Johns Hopkins University School of Medicine, Baltimore, MD, USA. [2]Institute for Cell Engineering, Johns Hopkins University School of Medicine, Baltimore, MD, USA. [3]Program in Emerging Infectious Diseases Duke-NUS Medical School, Singapore, Singapore. [4]Department of Microbiology, Universidad del Valle, Cali, Colombia. [5]Department of Biomedical Sciences, Colorado State University College of Veterinary and Biomedical Sciences, Fort Collins, CO, USA. [6]Department of Neurology, Johns Hopkins University School of Medicine, Baltimore, MD, USA. [7]Department of Medicine, Johns Hopkins University School of Medicine, Baltimore, MD, USA. [8]Department of Molecular Microbiology and Immunology, Johns Hopkins University Bloomberg School of Public Health, Baltimore, MD, USA. [9]Department of Biostatistics, Johns Hopkins University Bloomberg School of Public Health, Baltimore, MD, USA. [10]Department of International Health, Johns Hopkins University Bloomberg School of Public Health, Baltimore, MD, USA. ✉e-mail: hlarman1@jhmi.edu; mrobin85@jhmi.edu

protective antibody responses contribute to this increased risk in an immunopathological mechanism known as antibody dependent enhancement (ADE)[7]. However, some cross-reactive anti-flaviviral antibodies are cross-neutralizing and protective[8,9]. Cross-reactivity between flaviviruses thus has a multitude of public health implications, including for vaccine development[10]. The complexities associated with dengue and immunity to other flaviviruses also likely apply to less studied anti-arboviral cross-reactive antibody responses such as between alphaviruses[11].

Programmable phage display enables massively multiplexed analysis of antibody reactivities to libraries of peptides via Phage ImmunoPrecipitation Sequencing (PhIP-Seq)[12]. We and others have used PhIP-Seq to assess antibody responses to the human virome[13], comprehensively map antibody responses to coronaviruses[14,15], and deconvolute anti-viral antibody cross-reactivity[16]. Phage display has previously been applied to investigate anti-arbovirus antibodies to profile Zika infected individuals' antibodies against Zika[17], map the epitope of an anti-dengue NS1 antibody[18], profile dengue antibody responses in bats[19], and profile dengue antibody responses longitudinally in non-human primates inoculated with dengue[20]. However, with the exception of the pan-flavivirus phage library used to profile dengue antibodies in non-human primates[20], previous phage display platforms only include peptides covering reference strains of common human disease-causing arboviruses. Phage display has yet to be used to characterize viral strain or mutation specific effects on antibody specificity, used to map arbovirus cross-reactivity, or expanded to enable surveillance of zoonotic arboviruses from diverse genera.

We have generated a programmable phage display library that contains overlapping 56 amino acid peptides that represent the proteomes of 691 human and zoonotic arboviruses. PhIP-Seq with the pan-arbovirus phage library, ArboScan, enables detailed quantification of anti-arbovirus antibody reactivity with less than 1 microliter of serum or plasma, making it compatible with analysis of individual vector bloodmeals[12,21]. ArboScan can thus be used to measure antibodies to established and emerging arboviruses in humans, animal reservoirs, and arthropod vectors (Fig. 1a). We used ArboScan to detail antibody responses in an animal challenge study, in a Colombian cohort of Zika patients, and in a cohort of individuals challenged with dengue virus vaccine strains. ArboScan represents a new resource for characterizing human and animal arbovirus antibody responses at cohort scale.

## Results

### Design of ArboScan

The original VirScan PhIP-Seq library is comprised of 56 amino acid peptides that tile the proteomes of over 200 human viral species[13]. Only viral proteins that were part of a reference proteome, annotated as having human host, and demonstrated <90% sequence homology to another protein sequence were included. The resulting VirScan library therefore does not include most arboviruses, antigenic diversity within viral species, and viruses not known to infect humans. We sought to develop a phage library that could be used for worldwide surveillance of emerging arboviruses and to deeply characterize humoral immunity to well-characterized arboviruses. To this end, we identified all unique proteomes from viral genera thought to include arboviruses in GenBank as of August 2017, and divided these sequences into 56 amino acid peptides with 28 amino acid overlaps. Both ArboScan and VirScan quantify solely the fraction of anti-viral antibodies that bind the displayed 56-mer peptides, likely a minority of all anti-viral antibodies.

ArboScan greatly expands the diversity of arboviral antigens when compared to VirScan. ArboScan contains over 10-fold more alphavirus, flavivirus, orthobunyavirus, and phlebovirus peptides compared to VirScan, while adding previously absent viral genera, including orbivirus, nyavirus, and quaranjavirus. While VirScan contains peptides that tile the complete polyprotein of 15 flaviviruses, ArboScan contains peptides that tile the complete polyprotein of 98 flaviviruses (Fig. 1b), as well as eleven additional flaviviruses that lacked full genome sequences. While VirScan included peptides from 15 alphaviruses, ArboScan contains peptides tiling the full proteome of 35 alphaviruses and proteins from 3 partially sequenced alphaviruses.

### VARscore accurately characterizes targets of humoral immunity

To facilitate the interpretation of complex antibody profiles measured with ArboScan or VirScan, we developed an algorithm to identify viruses targeted by antibodies in a sample (see Methods). This algorithm integrates the intensity and breadth of antibody response to all peptides for each virus in a PhIP-Seq library to produce a Viral Aggregate Reactivity score (VARscore). It additionally adjusts for host, batch, and library factors which may confound comparison of PhIP-Seq results. To assess the performance of this approach, we applied the VARscore algorithm to VirScan data of 615 samples from healthy individuals from the United States prior to their participation in vaccine studies[22]. VARscores effectively distinguished viruses expected to be frequently targeted by antibodies from viruses expected to be less commonly targeted by antibodies in this population (Supplementary table 3, area under the receiver operator curve = 0.98 Fig. 1c, and area under the precision-recall curve = 0.97 Supplementary Fig. 2c). The viruses expected to be frequently positive intentionally include viruses that are not universally targeted such as cytomegalovirus (United States population seroprevalence of approximately 50%), while the viruses expected to be rarely positive were selected to include viruses such as hepatitis C virus which is expected to be positive at low frequency in the United States (-1–2%)[23]. Seroprevalence estimated by VARscores of respiratory syncytial virus A, Epstein-Barr virus, cytomegalovirus, hepatitis C, and HIV-1 are 94%, 89%, 47%, 1.8%, and 0.5% respectively (Supplementary Fig. 2d), largely matching with population seroprevalence in the United States[13,23,24]. VARscore represents an efficient unbiased metric for quantifying aggregate antibody reactivity to groups of peptides in a PhIP-Seq library and will be extended in the future to account for cross-reactivity.

### Arbovirus antibody profiling of animals and arthropod vector blood meals

Arbovirus life cycles require transmission via an arthropod vector and frequently involve non-human animal hosts. We therefore sought to evaluate the capacity for ArboScan to profile antibodies present in these contexts, beginning with vector bloodmeals. Public health programs have long relied on detection of live virus, nucleic acids, or antigens in mosquito samples in addition to changes in animal seroprevalence to surveil for arbovirus circulation[25]. We therefore determined whether anti-arboviral antibodies contained within arthropod blood meals were amenable to profiling via ArboScan. A standard serum antibody profile was compared to that derived from individual mosquito blood meals collected immediately, 24, and 48 h after feeding. The profile from an immediately harvested blood meal is indistinguishable from the standard serum profile (Concordance correlation coefficient (CCC) = 0.90, $p = 2.2 \times 10^{-16}$, Pearson's $R = 0.90$, $p = 2.2 \times 10^{-16}$). The arboviral targets of antibodies contained in a mosquito blood meal could still be measured by ArboScan 24 h post mosquito feeding, but blood meal antibody reactivities could no longer be effectively measured 48 h post feeding (24 h: CCC = 0.53, $p = 2.2 \times 10^{-16}$, Pearson's $R = 0.66$, $p = 2.2 \times 10^{-16}$, and 48 h: CCC = 0.135, $p = 0.0003$, Pearson's $R = 0.18$, $p = 0.0003$ respectively; Fig. 1d). These data establish the feasibility of using ArboScan to indirectly monitor arboviral exposures and immunity via analyses of arthropod vector blood meals.

We additionally applied ArboScan to profile antibodies from alpacas, goats, and horses challenged with Mayaro virus, Zika virus, or Japanese encephalitis virus, as well as 4 goat siblings, 2 of which had significant bluetongue virus neutralizing titer and 2 that did not

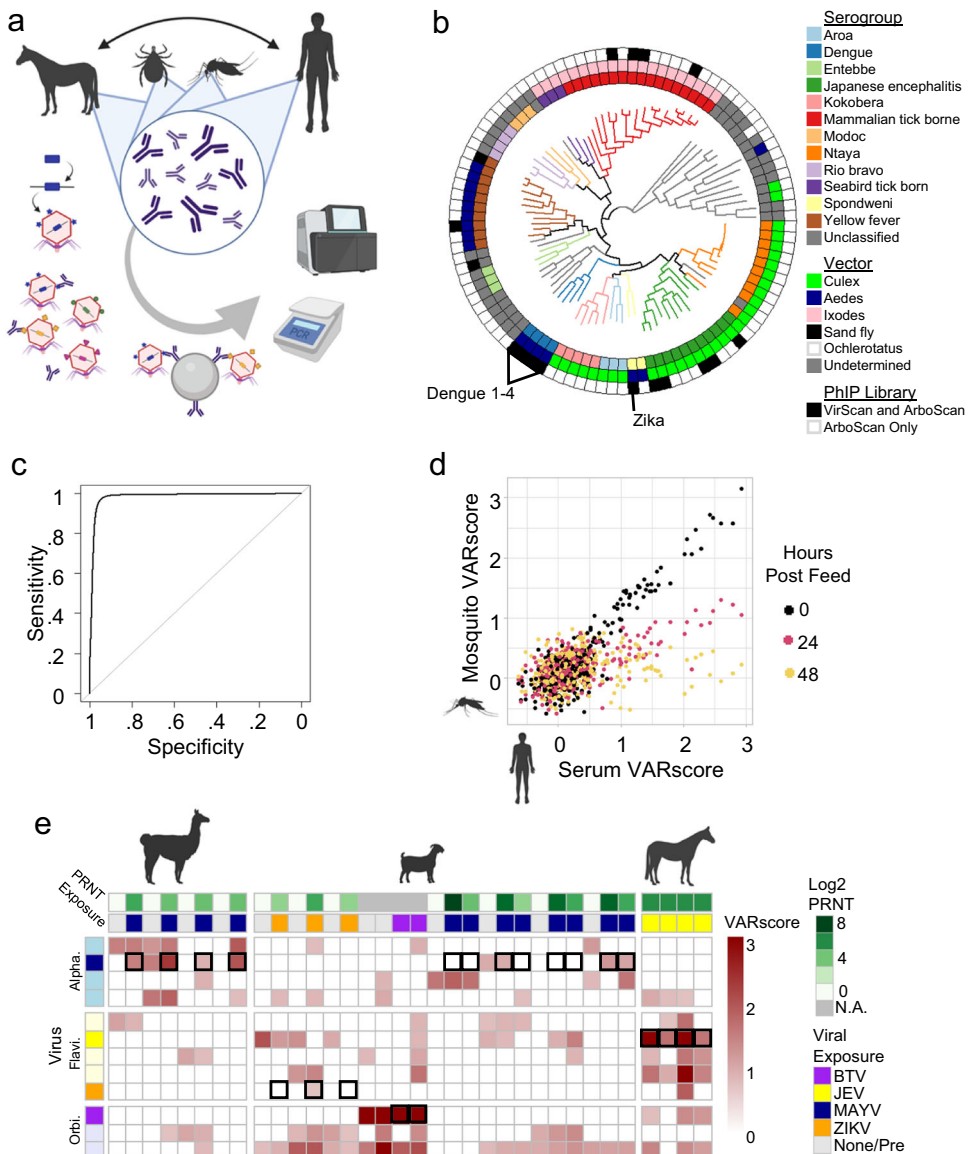

**Fig. 1 | IgG profiling with the T7 phage displayed ArboScan library. a** ArboScan enables comprehensive quantification of antibody responses to arboviruses in humans and non-human animals. **b** Phylogeny of 98 flaviviruses in ArboScan which had full length genomes or full polyprotein sequences available. 15 of these viruses were included in VirScan. 11 additional flaviviruses without full genome sequences have proteins included in ArboScan. **c** Receiver operator curve evaluating VARscore as a predictor to distinguish viruses with moderate or high expected seroprevalence from viruses with low expected seroprevalence (viruses listed in Supplementary Table 3) in healthy United States residents (AUROC = 0.98). The ROC was generated using a sliding VARscore cut-off. **d** VARscores from serum antiviral antibodies were compared to VARscores from anti-viral antibodies contained in processed mosquitoes harvested immediately post feeding, 24 h post feeding, and 48 h post feeding. **e** ArboScan was used to profile alpaca, goat, and horse antibodies before (Pre) and after arboviral exposure. VARscores for several alphaviruses, flaviviruses, and orbiviruses are shown. Boxes highlight the relevant virus with available PRNT data. Three alpacas and four goats were challenged with Mayaro virus (MAYV). Four horses were challenged with Japanese encephalitis virus (JEV). Three goats were challenged with Zika virus (ZIKV). **a**, **d**, and **e** were created with BioRender.com released under a Creative Commons Attribution-NonCommercial-NoDerivs 4.0 International license. Source data are provided as a Source Data file.

(Fig. 1e). Anti-arboviral antibody profiles of animal serum defined with ArboScan were unique and reproducible (Supplementary Fig. 3a). Pre-challenge animals had significant antibody responses to between 0.2% and 2.3% of viruses in ArboScan, while the two goat siblings without significant bluetongue virus neutralizing titer targeted 1.2% and 3.3% of the viruses in ArboScan (Supplementary Fig. 2e). Given that large study animals were housed in nonsterile environments, it is difficult to conclude with certainty that they have never been exposed to any arbovirus. As described previously in humans[16], anti-arboviral antibody profiles in animals were stable for at least 28 days (the maximum sampling period, Supplementary Fig. 3b). Of the four alpacas

challenged with Mayaro virus, three seroconverted via ArboScan by day 21, with the fourth alpaca displaying robust preexisting anti-Alphavirus antibody responses to multiple viruses including Mayaro virus (Fig. 1e). One of the four goats challenged with Mayaro virus seroconverted by day 14, and one other goat displayed pre-existing Mayaro virus antibodies (Fig. 1e). None of the goats challenged with Zika virus became viremic, but all three developed detectable neutralizing titers (PRNT 50 = 10, 10, and 40)[26]. Only the goat with the largest PRNT seroconverted by VARscore, indicating that the two other goats did not generate the depth and breadth of antibody reactivity required for a positive VARscore. All horses inoculated with Japanese

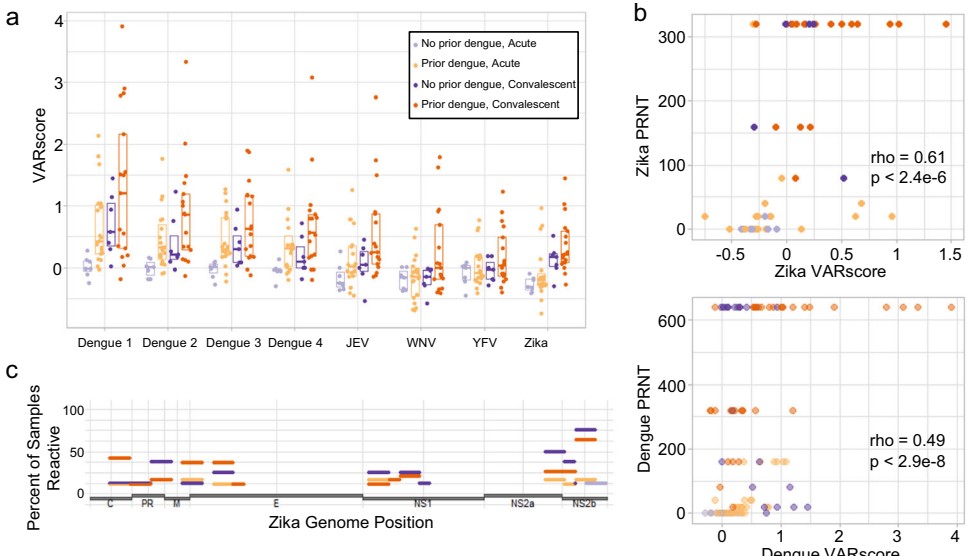

**Fig. 2 | ArboScan profiles of Zika infected individuals. a** Acute and convalescent serum was collected from individuals who tested positive for Zika RNA. VARscores from flaviviruses were compared between individuals who had pre-existing flavivirus IgG (prior dengue) and those who did not (no prior dengue) at acute and convalescent time points (No prior dengue, Acute $n = 8$; Prior dengue, Acute $n = 19$; No prior dengue, Convalescent $n = 8$; Prior dengue, Convalescent $n = 19$). Boxes indicate median and interquartile range of each group. **b** Zika VARscore were compared to Zika plaque reduction neutralizing titer (PRNT) (Spearman correlation, rho = 0.61, $p = 2.4 \times 10^{-6}$) and dengue VARscores were compared to corresponding dengue PRNT (Spearman correlation, rho = 0.49, $p = 2.9 \times 10^{-8}$). **c** Prevalence of antibody reactivity to Zika epitopes was evaluated for each sample group. Zika infected individuals with or without pre-existing flavivirus antibodies generated antibodies to Zika non-structural protein (NS) 2a, NS2b, membrane glycoprotein precursor (Pr), capsid (C), and envelope (E). Source data are provided as a Source Data file.

encephalitis virus displayed a robust antibody response (Fig. 1e). Among all the goats and horses profiled, consistent orbivirus antibody reactivity was detected. Although only two of the four goat siblings had significant bluetongue virus neutralizing titers by PRNT, all four of the goat siblings displayed robust bluetongue virus antibody reactivity by ArboScan. ArboScan is thus able to quantify disease-relevant antibody responses generated by recent arboviral infections in animal reservoirs. While ArboScan and the VARscore analysis pipeline are able to identify viral targets of strong polyclonal antibody responses in an unbiased fashion, they do not serve as a surrogate for highly sensitive and specific functional antibody characterizations for individual viruses like plaque reduction assays.

### Prior flavivirus infection shapes subsequent Zika antibody response

We next used ArboScan to evaluate arboviral antibody responses among patients who presented with a febrile illness in Colombia during the 2016 Zika epidemic[27]. Each study participant presented at enrollment with a positive Zika virus nucleic acid test. At initial and follow-up visits, serum was collected from patients and tested for Zika neutralization; half of the samples were additionally tested for dengue 1-4 neutralization. Prior to the emergence of Zika in 2016, dengue had been the only known endemic flavivirus in Colombia for nearly three decades, so presence of anti-Flavivirus IgG at presentation was taken to indicate prior dengue infection[28]. As expected, dengue was the most common target of pre-existing anti-flavivirus antibodies (Fig. 2a). Convalescent sera for patients who were flavivirus seropositive at presentation demonstrated increased reactivity not just to dengue but also West Nile virus and yellow fever virus (Fig. 2a) compared to patients who were flavivirus seronegative at presentation. This may be explained by secondary recall of antibody responses targeting epitopes conserved among flaviviruses. By contrast, individuals without evidence of prior dengue exposure usually responded to Zika virus infection by producing antibodies reactive only to dengue and Zika (Fig. 2a). In this case, the high degree of homology between Zika and

dengue viruses may explain the antibody reactivity to dengue[29]. We then evaluated the correlation of Zika VARscores with neutralizing activity. Indeed, Zika VARscores correlated with Zika PRNT (Spearman correlation, rho = 0.61, $p = 2.4 \times 10^{-6}$, Fig. 2b). In this cohort, neutralization of all dengue serotypes was correlated with each other (Supplementary Fig. 4) as well as with dengue VARscores (Spearman correlation, rho = 0.49, $p = 2.9 \times 10^{-8}$, Fig. 2b, each serotype shown separately in Supplementary Fig. 5a).

ArboScan additionally details epitope level antibody reactivities. In the acute setting, a minority of Zika virus infected individuals with prior dengue exposure had antibodies that bound Zika epitopes in capsid (C), envelope (E), nonstructural protein 1 (NS1), nonstructural protein 2b (NS2b), and nonstructural protein 5 (RNA-dependent RNA polymerase, NS5) (Fig. 2c). This could be due to preexisting cross-reactive antibodies or rapid recall of cross-reactive flavivirus antibodies to Zika infection. Zika infection with or without prior dengue infection generated antibody responses to C, Pr, membrane (M), E, NS1, NS2b, and NS5 (Fig. 2c). While Zika antibody responses targeting structural proteins and NS1 appear to cross-react with dengue, antibodies targeting NS2b specifically target Zika (Supplementary Fig. 5b). ArboScan data thus complement classic serology by revealing response characteristics that distinguish primary and secondary flavivirus infection.

### ArboScan distinguishes serotype-specific dengue antibody responses

Currently, low throughput PRNT testing is the standard method for differentiating antibody responses to dengue serotypes. We therefore asked whether neutralizing antibody responses to distinct dengue serotypes could be distinguished using ArboScan. To this end, we evaluated anti-arboviral antibodies in a cohort of individuals challenged with serotype-specific, recombinant dengue vaccine or challenge strains[30]. As with animal arboviral antibody profiles, human anti-arboviral antibody profiles were stable over the entire sampling period (up to 120 days, Supplementary Fig. 3c). Challenge with dengue virus

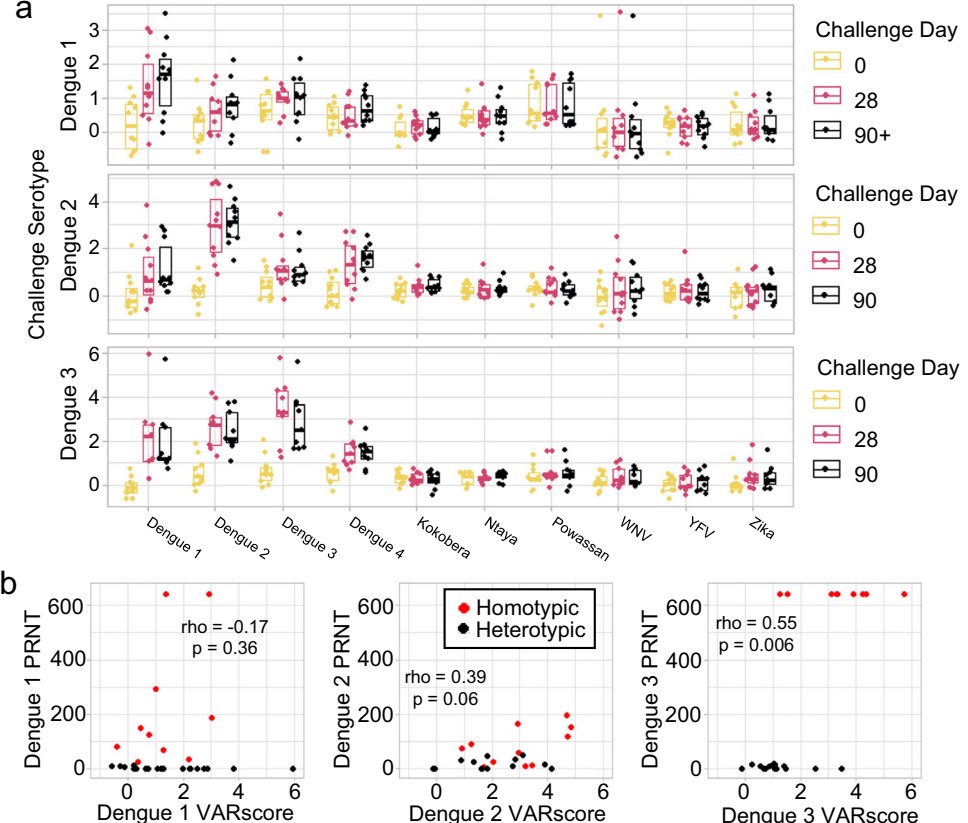

**Fig. 3 | ArboScan profiles of individuals challenged with dengue. a** Longitudinal VARscores for dengue serotypes and representative flaviviruses for individuals challenged with single serotype dengue challenge strains (Dengue 1: day 0 $n = 10$, day 28 $n = 10$, day 90+ $n = 10$; Dengue 2: day 0 $n = 11$, day 28 $n = 11$, day 90 $n = 10$; Dengue 3: day 0 $n = 10$, day 28 $n = 9$, day 90 $n = 9$). Boxes indicate median and interquartile range of each group. **b** Dengue VARscores were compared to corresponding dengue plaque reduction neutralizing titer (PRNT). Dengue 3 VARscores correlate with dengue 3 neutralization (Spearman correlation, rho = 0.55, $p = 0.006$). Dengue 1 and dengue 2 VARscores do not correlate with dengue 1 and dengue 2 neutralization (Spearman correlation, rho = −0.17 and 0.39 respectively, $p = 0.36$ and 0.06 respectively) because (i) challenge with heterologous dengue serotypes produced increases in VARscores but not in neutralization and (ii) challenge with dengue 1 and dengue 2 induced variable immune responses. Source data are provided as a Source Data file.

vaccine strains generated an increase in antibodies to the corresponding dengue serotype in the majority of cases at 28 and 90 days (Supplementary Fig. 6a). The dengue 1 vaccine strain produced the weakest and least consistent antibody response, and several individuals failed to mount an antibody response post-challenge (Supplementary Fig. 6a). Dengue challenge often produced an increase in antibody binding to heterologous dengue serotypes, but the greatest increase in antibody reactivity most often corresponded to the challenge serotype (Fig. 3a).

Inclusion in this dengue challenge study required negative preexisting flavivirus antibodies by negative $PRNT_{50}$ assay to all 4 dengue stains, yellow fever virus, St. Louis encephalitis virus, West Nile virus, and Zika (after the 2015 Zika outbreak). In this setting, we observed minimal induction of heterotypic antibody reactivity to other flaviviruses following dengue challenge. However, ArboScan identified several individuals with significant preexisting antibody reactivity to Powassan virus. Additionally, one individual had consistently high antibody reactivity to West Nile virus in discordance with having a negative West Nile virus PRNT assay during screening (Fig. 3a). This may be either due to the presence of non-neutralizing antibodies detected by Arboscan or due to a very recent infection between screening and challenge.

To evaluate the functionality of cross-reactive anti-dengue antibodies generated by dengue challenge, we compared ArboScan antibody profiles against gold standard PRNT data for each dengue serotype. In this dengue naïve cohort, neutralization of different

dengue serotypes was not correlated because each challenge produced a specific increase in PRNT to a single serotype. Even though challenge usually generated cross-reactive but not cross-neutralizing antibodies, dengue 3 serotype-specific VARscores correlated with PRNT across all challenges (Spearman correlation, rho = 0.55, $p = 0.006$, Fig. 3b). Challenge with dengue 2 and dengue 3 produced large increases in dengue 1 and dengue 4 VARscores but not PRNT, and further investigation is needed to evaluate potential links between these cross-reactive antibodies, ADE, and the risk of developing severe dengue disease.

## Serotype-specific and cross-reactive dengue epitopes defined with ArboScan

We then characterized the evolution of epitope specificities targeted by antibodies post dengue challenge. Several epitopes were frequently targeted by preexisting antibodies in dengue naïve individuals, including E and Pr for dengue 4 and M for both dengue 3 and dengue 4 (Supplementary Fig. 7). Dengue challenge did not result in more frequent antibody reactivity to these epitopes suggesting that these peptide reactivities are due to off-target cross-reactivity as opposed to flavivirus-directed cross-reactivity. The weakly immunogenic dengue 1 challenge strain elicited antibodies reactive to C, Pr, multiple epitopes in E, and NS3 (Supplementary Fig. 7, Supplementary Fig. 6b). Following challenge with dengue 2, >75% of individuals developed antibodies reactive to dengue 2 C, Pr, M, and the N terminus of E. Many individuals' antibodies additionally reacted with homologous peptides from

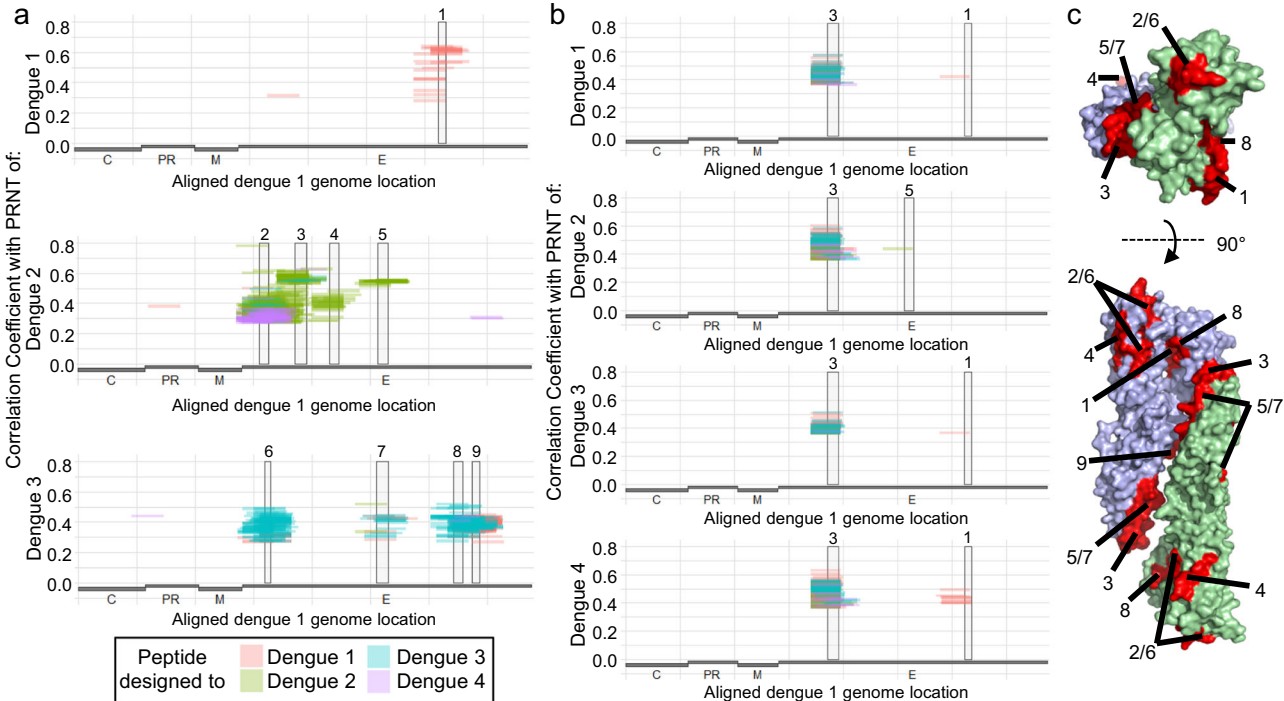

**Fig. 4 | Correlations between dengue envelope epitope reactivities and dengue neutralization. a** Antibody reactivities to all dengue peptides were evaluated for correlation with dengue neutralization in the dengue challenge cohort. The vast majority of peptides that significantly correlated with dengue plaque reduction neutralizing titer (PRNT) were in envelope (E). Minimal epitopes that may explain E peptides were as follows: epitope 1: AAs 342-355, epitope 2: AAs 31-46, epitope 3: AAs 91-111, epitope 4: AAs 151–165, epitope 5: AAs 235-251, epitope 6: AAs 39-48, epitope 7: AAs 232-251, epitope 8: AAs 363-378, and epitope 9: AAs 394-407.
**b** Antibody reactivity to dengue peptides that correlated with dengue neutralization in the Colombia Zika cohort. Epitopes 1, 3, and 5 were transferred from **a**.
**c** Mature dengue envelope dimer with epitopes that correlate with neutralization highlighted in red. Source data are provided as a Source Data file.

other dengue viruses (Supplementary Fig. 7). Dengue 3 challenge resulted in the most robust antibody responses; virtually all regions of C, Pr, M, E, NS1, NS2a, and NS3 were targeted by antibodies in most individuals following dengue 3 challenge (Supplementary Fig. 7). Epitope breadth and strength of the fraction of antibody reactivities measured with ArboScan corresponded well to other measures of antibody response strength.

### ArboScan identifies antibody specificities that correlate with dengue neutralization

To define particular epitope reactivities that correlate with serotype neutralization, we calculated the Pearson and Spearman correlation coefficients between each dengue peptide in ArboScan with dengue 1, 2, and 3 PRNT (dengue cohort–Supplementary Fig. 8a–h, Colombia cohort–Supplementary Fig. 8i–p). A minimum set of epitopes explaining these correlations were then determined as described in the methods (Fig. 4a). In the dengue challenge cohort, one epitope (E 342-355) correlated with dengue 1 neutralization. Four epitopes (E 31-46, 91-111, 151-165, and 235-251) correlated with dengue 2 neutralization. Four epitopes (E 39-50, 232-251, 363-378, and 394-407) correlated with dengue 3 neutralization. Epitope consensus sequences were generated from all overlapping correlating peptides and compared to dengue 1–4 reference sequences (Supplementary Fig. 9).

We proceeded to evaluate epitope reactivities that correlate with dengue neutralization in the Colombian cohort. In this cohort, reactivity to peptides that overlap with epitope 3 correlated with neutralization of all 4 dengue serotypes (Fig. 4b). This epitope, fusion loop, may be bound by antibodies that have beneficial or detrimental function. Fusion loop has been described as the target of a cross-reactive dengue antibody that protected mice from Zika infection[9], the target of antibodies that neutralized all dengue serotypes[31], and the target of antibodies that mediate ADE[32,33]. We

then evaluated the breadth of antibody cross-reactivity to fusion loop across all known flaviviruses (Fig. 5a). Every individual had measurable antibody reactivity to fusion loop following Zika infection, as did 16 of 19 individuals with pre-existing flavivirus antibodies. Breadth of antibody cross-reactivity to fusion loop ranged from dengue 3 specific to broadly cross-reacting with fusion loop of flaviviruses from all described serogroups. Multiple sequence alignments of fusion loop peptides bound by four example sera with differing degrees of cross-reactivity highlight binding modalities of specific and cross-reactive fusion loop antibody responses (Supplementary Fig. 10). The core DRGWGNGCGLFGKG motif (dengue-1 E 97-111) is completely conserved across almost all flaviviruses transmitted by mosquitoes, including dengue, Zika, West Nile, Japanese encephalitis, and yellow fever viruses. In addition to the dengue fusion loop antibody reactivity correlating with dengue neutralization, Zika fusion loop antibody reactivity was associated with Zika neutralization (two-sided fisher's exact test, $p = .005$, Fig. 5b). Unbiased investigation using ArboScan highlighted flavivirus epitopes that associate with neutralization and may rapidly identify candidate epitopes with potential functional importance from less well-studied arboviruses.

## Discussion

In this study, we present ArboScan, a programmable phage display library for detecting anti-arboviral antibody responses in high throughput and at epitope level. We show that ArboScan may be employed for arbovirus surveillance in animal reservoirs, including indirectly via testing antibodies contained within arthropod vector blood meals which may facilitate surveillance of difficult to sample animal populations[34]. ArboScan was used to identify epitope reactivities that correlate with neutralization, the strongest of which have been validated in orthogonal studies.

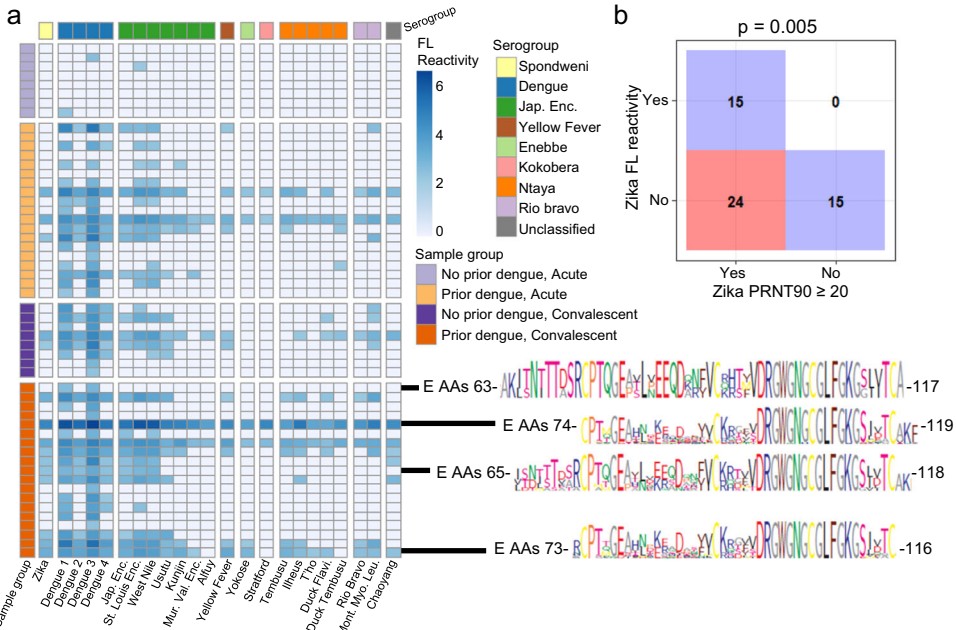

**Fig. 5 | Breadth of fusion loop reactivity.** Antibody reactivity to fusion loop (FL) was evaluated for every flavivirus in ArboScan for all samples from the Colombia cohort. Only viruses with ≥3 reactive samples were included for visualization. Epitope icons were included for four sera of varying degrees of cross-reactivity. Jap. Enc. Japanese encephalitis, Mur. Val. Murray Valley, Duck Flavi. Duck Flavivirus, Mont. Myo. Leu. Montana Myotis Leukoencephalitis virus. **b** Confusion matrix of each individual by detectable Zika fusion loop antibodies and Zika neutralizing antibodies (two-sided Fisher's exact test, $p = 0.005$). Source data are provided as a Source Data file.

Technology harnessing next-generation sequencing has advanced public health surveillance[35]. Despite these advances, genetic material must be abundant in individuals at the time of sampling for metagenomic approaches to be successful. Surveillance methods that rely on acute illness sampling must be applied to large populations to identify uncommon diseases. In contrast, serologic methods retrospectively interrogate years of exposure in a single sample. As such, VirScan has increasingly been applied to determine the etiology of syndromes that remain unexplained after exhausting all other modalities of investigation[36,37]. With ArboScan we greatly expand the scope of PhIP-Seq to detect viruses that may not currently be common human pathogens, yet form the pool from which future pandemics may emerge. One such example is Zika virus—which was not included in the original VirScan library given its obscurity prior to its emergence in the Americas in 2015.

In order to derive single virus scores from ArboScan data, we developed a viral aggregate reactivity score algorithm, VARscore, that assesses the signal to noise of virus level antibody reactivity within each individual profile. This score was used to identify targets of antibody responses without prior knowledge of a virus's immunodominant epitopes or requiring the use of seropositive and seronegative controls. The VARscore approach thus enables quantification of antibody reactivity in different species, expected to have unique antibody responses, and to less-studied viruses that may lack extensive serologic characterization. We anticipate VARscores will be most useful in settings where cross-sectional or longitudinal unbiased seroprevalence is of interest. ArboScan may eventually find utility as a clinical diagnostic, but this will require an improved approach to deconvolution of cross-reactivity, larger scale testing with well characterized samples, and use of paired acute and convalescent samples. We do not present data to support that ArboScan is sufficiently sensitive or specific for diagnosis of specific arbovirus species in individual patients.

We used ArboScan to correlate antibody reactivity to several dengue epitopes with dengue neutralization. Several residues in epitope 4 have been shown to be critical for certain rheses macaque neutralizing antibodies[38] as well as human neutralizing antibodies[8], and epitopes 5 and 7 contain another key residue for binding of these human antibodies[8]. In a mature folded envelope, epitopes 5 and 7 are adjacent and form part of the surface of D1. Epitope 3, an immunodominant linear stretch of the fusion loop, was found to be associated with neutralization in two independent cohorts. Antibodies that bind fusion loop may have either beneficial or detrimental properties, as they have also been associated with ADE[32,33,39]. The quantity of antibodies that bind fusion loop may determine their function, since anti-fusion loop antibodies that are neutralizing at high concentrations may promote ADE at lower concentrations[32,33]. Nonetheless, this epitope has been described as the binding site of broadly neutralizing antibodies with functionality against diverse flaviviruses[8,9,31,40,41], supporting a functional role for antibodies that bind fusion loop. Multiple antibodies with nonlinear epitopes have been described as binding both fusion loop and the DII bc loop[31,42]; ArboScan peptides that have the fusion loop at the C terminus additionally contain the DII bc loop. Peptides with both fusion loop and DII bc loop were much more likely to correlate with dengue neutralization in the Colombia Zika cohort as compared to peptides with fusion loop at the N terminus. This potentially indicates that some individuals in this cohort generate neutralizing antibodies that bind to an epitope formed by both fusion loop and bc loop, and ArboScan peptides containing both these regions adopts a nonlinear structure to present a similar epitope. Future investigations may yield additional insights into the specific binding modalities that distinguish protective vs infection enhancing anti-fusion loop antibodies.

Our study has several important limitations. First, the animal challenges were performed with animals not known to be reservoirs of challenge viruses. Challenge infections frequently did not induce high viremia and strong antibody responses. Second, the dengue challenges were performed with attenuated virus which may result in an immune response different from those caused by infection with naturally occuring virus. Third, while determining virus level targets of

antibody responses, we did not account for antibody cross-reactivity. Deconvoluting anti-viral antibody cross-reactivity would be critical in defining viral exposures based on anti-viral antibody profiles. Fourth, our antibody profiling focused on IgG and will have missed IgM antibody reactivity in the acute setting. Profiling IgM with ArboScan represents a useful avenue of investigation for identifying acute infection and estimating arbovirus exposure incidence. Finally, PhIP-Seq with 56-mer peptide libraries principally detects the fraction of antibodies that bind to linear epitopes or epitopes completely contained within a displayed peptide, a small fraction of an antibody response induced by viral exposure. Our approach will not detect antibodies to highly conformational or discontinuous epitopes or epitopes containing post-translational modifications. ArboScan therefore does not identify all of the antibody specificities targeted in response to an arbovirus infection. Nevertheless, the identification of a minority of antibody reactivities to nearly every arbovirus may serve as a useful tool for epidemiologists seeking to broadly surveil for arbovirus exposure in diverse global populations, to vaccinologists interested in selecting potential epitope candidates for future study, and to virologists generating hypotheses about antibody responses across related viral species. Future platforms may complement ArboScan by displaying known conformational epitopes, protein domains, and full-length proteins to increase the diversity of reactivities measured in a single antibody profiling assay.

The limitations on reactivity detection by ArboScan additionally apply to the identification of epitopes of neutralizing antibodies, the majority of which are conformationally dependent and/or dependent on post-translational modifications. ArboScan can therefore only be used to map the small subset of neutralizing antibodies that bind epitopes entirely contained within the displayed 56-mer peptides. Furthermore, correlating neutralizing titer with epitope level reactivities is insufficient to prove that all antibodies that bind a particular epitope are directly neutralizing. Nevertheless, we have previously used a similar approach to identify several epitopes as likely to be functionally important for SARS-CoV-2[14], which were subsequently confirmed to be the target of neutralizing antibodies[43–45], highlighting the utility of this approach. ArboScan could be used in the future to rapidly identify epitopes of interest for monoclonal therapeutic or vaccine development for new or emerging viruses.

ArboScan fills an important niche in global health surveillance – a means to efficiently surveil for antibodies to all arboviruses in people and reservoirs without the complexity and cost of deploying hundreds of individual assays.

## Methods
### Specimens
**Cohort 1—Veterinary.** Samples were collected as part of previous studies[26]. The three goats challenged with Zika virus, the four alpacas challenged with Mayaro virus, and the four goats challenged with Mayaro virus each received 100,000 plaque forming units (PFU) per animal subcutaneously. The four horses received attenuated Japanese encephalitis virus as part of an ongoing vaccine trial. The goats with positive and negative bluetongue virus neutralizing titers were not intentionally exposed and were cohabitating.

Most animals were housed in the animal biosafety level 3 facility at Colorado State University for the duration of the study and all procedures were approved by the Colorado State University Institutional Animal Care and Use Committee. The goats evaluated for bluetongue virus neutralization were siblings from a local farm.

**Cohort 2—Colombian Zika outbreak.** During the 2016 Zika outbreak in Colombia, individuals were referred to the Universidad del Valle Virology Laboratory, Cali, Colombia, by attending clinicians at different health centers. Zika virus infection was confirmed by qRT-PCR in blood and urine specimens from individuals with suspected Zika

disease[46,47]. A subset of these Zika cases are part of another study[27]. The study protocol was approved by the institutional review boards at Universidad del Valle (protocol 034-016, IRB Approval # 006-016) and the Johns Hopkins University School of Medicine (IRB 00093149), and written consent was obtained from all patients.

**Cohort 3—Dengue human challenge.** Participants enrolled in clinical trials for dengue vaccine candidates were challenged with recombinant live attenuated dengue virus of either serotypes 1 (ClinicalTrials.gov identifier NCT02392325 and NCT00473135), 2 (NCT02317900), or 3 (NCT02873260)[30]. Participants were enrolled in the United States and were screened for previous flavivirus exposure by history and by assaying for neutralizing antibodies (plaque reduction neutralizing antibody titer 50%, $PRNT_{50}$) to dengue 1, dengue 2, dengue 3, dengue 4, yellow fever virus, Saint Louis encephalitis virus, West Nile virus, and after 2015, Zika virus. A $PRNT_{50} < 1{:}10$ was considered seronaive. Participants were challenged with recombinant dengue viruses that were modified from naturally occurring dengue strains with 30 or 31 nucleotide deletions in the 3' untranslated region (UTR). The challenge viruses were rDEN1Δ30−dengue 1 Western Pacific, rDEN2Δ30 - dengue 2 Tonga/74, and rDEN3Δ30−dengue 3 Sleman/78. The described trials were sponsored by the National Institute of Allergy and Infectious Diseases and protocols were approved by IRBs at the University of Vermont and Johns Hopkins Bloomberg School of Public Health where the studies were conducted.

**Cohort 4—Vaccine Research Cohort (VRC).** Healthy volunteers recruited for various vaccine trials at the National Institutes of Health provided samples prior to participation in vaccine trials. VirScan was performed on these samples[22].

**Mosquito blood meals.** A healthy volunteer donated a blood sample. Aliquots of this blood was then provided to mosquitos via a feeder and individual mosquitos were subsequently harvested. Harvested mosquitos were crushed in 100 uL PBS and then treated identically to serum samples.

### ArboScan library generation
The ArboScan phage display library incorporates 691 arbovirus species' genomes comprising every listed arbovirus entry in GenBank in 2017, as well as 5 control viruses, for a total of 25,138 unique protein sequences. Selection of viruses for inclusion started with literature review of ArboCat (https://wwwn.cdc.gov/arbocat/), International Committee on Taxonomy of Viruses (https://ictv.global/), and other sources for viral genera that included at least one viral species suspected of transmission via an arthropod vector. Proteomes for possible arboviruses were identified by searching GenBank for viruses included in the following viral genera or taxonomies: Coltivirus (txid10911), Nyavirus (txid1513295), Flavivirus (txid11051), Alphavirus (txid11019), Nairovirus (txid1980517), Quaranjavirus (txid1299308), Orbivirus (txid10892), Orthobunyavirus (txid11572), Phlebovirus (txid11584), Thogotovirus (txid35323), unclassified peribunyaviridae (txid39718), Ephemerovirus (txid32613), Vesiculovirus (txid11271), Almendravirus (txid1972682), Curiovirus (txid1985688), Hapavirus (txid1972611), Ledantevirus (txid1978532), Tibrovirus (txid1299306), Tupavirus (txid1513300), and unclassified Rhabdoviridae (txid686606 and txid35303). Unique polypeptide sequences were divided into 56 amino acid peptides with 28 amino acid overlaps, and exact duplicate peptides were removed. Peptides with less than three amino acid differences were additionally filtered for a subset of viruses, including 11 viruses known only to infect arthropods and 7 highly represented viruses of lower perceived human health priority (listed in Supplementary table 1). The final library contained 210,976 peptides (breakdown by genus in Supplementary table 2). Oligonucleotides encoding these peptides were

then synthesized by Twist Bioscience and cloned into a mid-copy T7 bacteriophage display system as performed previously[21]. Phage libraries were prepared as previously described[21]. Sequencing the expanded ArboScan bacteriophage library showed that 98% of the designed peptides were successfully cloned (Supplementary Fig. 1).

## PhIP-Seq
Phage ImmunoPrecipitation Sequencing (PhIP-Seq) utilizing the VirScan or ArboScan peptide library was performed as previously described. Briefly, 0.2 μl of serum sample was incubated with the phage library overnight. Serum antibodies and antibody-bound phage were then immunoprecipitated with protein A and protein G coated magnetic beads. The genetic material of the antibody-bound phage then underwent PCR amplification and sample barcoding before next-generation sequencing on an Illumina NextSeq500.

## Neutralization tests and clinical ELISAs
Neutralizing titers and additional clinical serology for each study were collected separately for each study cohort as follows:

**Cohort 1—Veterinary cohorts.** The production of neutralizing antibodies to challenge viruses was determined by plaque reduction neutralization test (PRNT). Briefly, serum was first heat-inactivated for 30 min in a 56 °C water bath. Then serum samples were diluted two-fold in BA-1 media starting at a 1:5 dilution on a 96-well plate. An equal volume of corresponding virus was added to the serum dilutions and the sample-virus mixture was gently mixed. The plates were incubated for 1 h at 37 °C. Following incubation, serum-virus mixtures were plated onto confluent Vero plates. Antibody titers were recorded as the reciprocal of the highest dilution in which > 80% of virus was neutralized.

**Cohort 2—Colombia Zika outbreak.** PRNTs for Zika virus and dengue 1-4 serotypes were performed following previously established protocols[48,49]. In short, PRNT was performed in Vero cell monolayers at 90% confluency seeded in 24-well plates. Serum specimens were heat-inactivated, serially 4-fold diluted (1:20-1:320) and mixed at 1:1 volume of a viral suspension that produces 80–100 plaque-forming units (PFU) of each challenge virus per well and incubated at 37 °C for 1 h. Virus strains used in the assay were Zika virus isolated in Cali, Colombia and prototype laboratory dengue 1–4 strains. The virus-serum mixtures were then inoculated onto Vero cells (three replicates) to allow 1-h virus adsorption at 37 °C prior to overlay with a semisolid medium (1.2% carboxymethylcellulose, 50% 2X MEM, 1% non-essential amino acids, 2% FBS, 1% penicillin/streptomycin and 1% amphotericin B) and incubated for 6 days at 37 °C, 5% $CO_2$ to allow virus plaques to develop. Infected cells were next fixed with 20% formaldehyde solution and stained with crystal violet to count plaques. Neutralizing titers were determined by PRNT with a cutoff value of 90% reduction in plaque counts according to Lanciotti RS, et al.[47].

The presence of flavivirus cross-reactive antibodies was examined by dengue IgM-capture and IgG-capture enzyme-linked immunosorbent assays (ELISAs) from PANBIO Diagnostics ®, Australia (Ref 01PE20 and Ref 01PE10) in acute phase (<10 days) and convalescent phase (≥10 days) paired-serum specimens, following the manufacturer's guidelines. Zika virus and dengue antibody reactivity were indistinguishable with the commercial ELISA tests available[47,50]. A positive IgG test result at acute disease phase was interpreted as a marker of previous dengue exposure, because the assay cutoff was set to detect the increased IgG titers characteristic of an anamnestic antibody response in dengue secondary infections[28].

**Cohort 3—Dengue human challenge.** Levels of neutralizing antibodies to dengue 1, dengue 2, dengue 3, and dengue 4 were measured by plaque reduction assay, as described previously[51],

analogous to how PRNTs were performed on the Colombia Zika outbreak cohort.

## Informatics
Analysis was performed in R (v 4.2.2). The output from short-read sequencing of the immunoprecipitated phage libraries underwent initial processing as previously described[52], with the addition that a pseudo count was added for every peptide to control fold-change estimates of lowly abundant library members. Sequencing reads attributed to phage library members were counted via exact matching. The magnitude of antibody binding to each peptide was determined using the EdgeR package (v 3.32.0) in R[53]. All other analysis of PhIP-Seq data was performed in R subsequent to this initial processing.

Analyses of biological sequences were performed with R packages Biostrings (v 2.66.0) and msa (v 1.28.0). Visualizations were made with R packages ggplot2 (v 3.4.0), ggtree (v 3.4.4), circlize (v 0.4.15), pheatmap (v 1.0.12).

## Viral Aggregate Reactivity Score (VARscore)
Interpreting the antibody response to hundreds of thousands of individual peptides that comprise a PhIP-Seq library to adjudicate whether a serum sample demonstrates exposure to a specific virus is challenging. Contributors to this challenge include unequal library representation by virus, batch effect, cross-reactivity, and heterogeneity by individual in the target, breadth, and intensity of antiviral antibody responses. Previous methods used to identify targets of antibody responses with PhIP-Seq relied on prior knowledge of a virus's immunodominant epitopes or uniform tiling of viruses as in VirScan[13]. To create a more general method for evaluating serum antibody reactivity to each virus represented by a phage library that incorporates both the breadth and intensity of antibody response while adjusting for unequal library representation, an aggregate reactivity score algorithm was developed. The algorithm was run on each sample individually. Viral aggregate reactivity scores (VARscores) were only calculated for viruses that were represented by ≥ 50 peptides in the library (570 viruses). An intermediate reactivity metric (r) was calculated for each virus (v) as shown in Eq. (1):

$$r_v = \sum_{i=1}^{n} \frac{\log_2 f_i}{n} ; peptide_i \in Virus_v \tag{1}$$

where $f_i$ is the fold change for the $i^{th}$ peptide $p_i$ that was designed to virus v. Fold changes were floored at 1. This reactivity metric is then compared to distributions generated by randomly selecting n random peptides 1000 times from a sample's reactivity profile. Gamma distributions were empirically determined to best fit the distributions generated by randomly selecting peptides (Supplementary Fig. 2a); the parameters of a best-fit gamma distribution change regularly with the number of peptides selected (Supplementary Fig. 2b). To expedite computation, random distributions were created for $n = 30, 60, 120, 240, 480, 960, 1960, 3920, 7840$, and 15680, and gamma distribution parameters for intermediate values of n were determined with a linear regression model. A p-value was defined by integrating the tail of a gamma distribution with appropriate parameters greater than $r_v$. An effect size—VARscore—was defined from a similarly calculated p-value but using a gamma distribution with variance multiplied by the square root of the number of peptides (n) and with an identical mean. This p-value was then converted to the z-score that would have produced that p-value from a normal distribution. As the distributions generated from random peptides were intended to represent background reactivity to viral peptides, this algorithm was run iteratively (a maximum of 10 times) with each subsequent iteration excluding peptides from random selection if they derived from a "positive virus". The internal VARscore cutoff for positivity was determined empirically from the VRC cohort (Fig. 1c, Supplementary Fig. 2c), and the internal

$p$-value cutoff was determined with a Bonferroni correction controlling the family wide error rate at 0.05 for each individual. As every sample is run independently, this algorithm is highly parallelizable.

## Visualization of antibody reactivities mapped by genomic location

ArboScan includes peptides that tile the proteome of many dengue virus and Zika virus lineages. To visualize immunodominant epitopes, each dengue virus peptide was aligned to the polyprotein from dengue virus reference genomes. Similarly, Zika virus peptides were aligned to the Zika virus reference polyprotein. The proportion of individuals in a given sample group with reactivity to each peptide was calculated. Then, for every amino acid location in a viral polyprotein, the largest proportion of individuals with reactivity to a peptide that overlaps that location was determined and plotted.

## Correlating epitopes with neutralization

To define epitope reactivities that correlate with dengue neutralization, overlapping peptides were not collapsed. For the dengue challenge cohort, correlations were calculated using all day 0 and day 28 reactivity profiles and were defined as the correlation between dengue PRNT and the fold change value for a given peptide. All day 0 PRNTs were set to 0, and all day 28 PRNTs were set to the maximum measured PRNT post-challenge. Each dengue peptide was evaluated for correlation with neutralization of dengue serotypes 1, 2, and 3. Peptide tiles were considered for further analysis if both Pearson and Spearman correlation $p$-values were < 0.05. All peptides were mapped to the dengue-1 reference genome, and where three or more peptides overlapped, a minimal epitope was determined that could explain the peptide reactivities. For the Colombian cohort, all samples where dengue PRNTs were measured were included, and each peptide was correlated with dengue 1, 2, 3, and 4 neutralization.

## Anti-fusion loop antibody response breadth and binding motifs

Potential fusion loop containing flavivirus peptides were identified by aligning all flavivirus peptides to each of the dengue fusion loop peptides that correlated with dengue neutralization in the Colombia cohort. Alignments were performed with mmseqs2[54], and peptides that aligned to a dengue fusion loop peptide with an $e$-value $<1 \times 10^{-5}$ were included for further analysis. The maximum fold change from all fusion loop peptides designed to a given virus was used for fusion loop heatmaps, and viruses were only plotted if they reacted with more than 3 samples. Binding motifs were generated for four samples by aligning the maximally reactive fusion loop peptide from each reactive virus.

## Flavivirus and alphavirus phylogeny

To generate phylogenies, available full genomes or proteomes for flaviviruses were downloaded from GenBank in August 2022. Phylogenies were created using MEGA X[55]. Serogroup and vector annotations were derived from previous literature[4,56–58].

## Statistics and reproducibility

No statistical method was used to predetermine sample size. No data were excluded from the analyses. Sample locations were randomized in 96-well plates and sample identities were blinded for PhIP-Seq experiments. All statistical tests were two sided.

## Reporting summary

Further information on research design is available in the Nature Portfolio Reporting Summary linked to this article.

## Data availability

Source data are provided with this paper. PhIP-Seq data were deposited into the Sequence Read Archive database under accession number PRJNA1106197 and are available at the following URL: https://www.ncbi.nlm.nih.gov/bioproject/1106197 Additional raw and processed data generated for this study are freely available from authors on request. Source data are provided with this paper.

## Code availability

The ARscore package for generating aggregate reactivity scores, implemented in R v 4.2.2, is available on GitHub at https://github.com/wmorgen1/ARscore.

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

## Acknowledgements

C.P., B.P. and H.B.L are supported by NIH R01 NS110122. B.P. is supported by a Fulbright Visiting Research Scholarship. H.B.L. and W.R.M are supported by NIH R01 GM136724. M.R. and H.B.L are supported by a Johns Hopkins Catalyst Award. The work conducted at Duke-NUS Medical School was partially funded by a grant from the National Medical Research Council (L-FW, OFLCG19May-0034). D.Z. is supported by NIH T32 AI007291.

## Author contributions

All authors reviewed and contributed to the preparation of the manuscript. Additional author roles are noted below. W.R.M designed the study, performed testing, analyzed data, wrote the manuscript. W.N.C performed experiments. B.P. collected samples and performed testing. D.R.M. provided bioinformatics support. I.R. collected samples, consulted with data analysis. C.A.P collected samples. R.B. collected samples. D.Z. analyzed data. D.N. consulted on study design. I.R. provided statistical and bioinformatics expertise and analyzed data. A.D. collected samples and consulted on study design. L.F.W. designed experiments. H.B.L. designed the study and analyzed data. M.R. designed the study and analyzed data.

## Competing interests

H.B.L. is an inventor on an issued patent (US20160320406A) filed by Brigham and Women's Hospital that covers the use of the VirScan technology, and is a founder of Infinity Bio, Portal Bioscience and Alchemab. The remaining authors declare no competing interests.
