## [Peer Review File · Nature Communications]

Precision Arbovirus Serology with a Pan-Arbovirus PeptidomeEditorial Note: This manuscript has been previously reviewed at another journal that is not operating a transparent peer review scheme. This document only contains reviewer comments and rebuttal letters for versions considered at *Nature Communications*.

REVIEWERS' COMMENTS

Reviewer #1 (Remarks to the Author):

The submission is a revised version where the focus of the manuscript is on the use of an extended peptide phage display library generated from human and zoonotic arboviruses on a T7 phage platform which is named ArboScan. The system was applied against Zika patient materials to identify unique antibody signatures for the specific infection. Additionally, a tailor made algorithm was developed to sieve the peptide data generated to identify key residues associated with the infection post sequencing. The depth of analysis and sample cohorts used is greatly appreciated as it allows for an in depth analysis and comparison.

The authors addressed the comments well.

Reviewer #2 (Remarks to the Author):

Morgenlander et al, phage display approach of 56-mer peptides for Arbovirus family to define the serological fingerprint of antibody reactivity in animals and humans. Furthermore, they correlated some epitope reactivity with functional neutralization activity generated by these viruses. The authors have revised the manuscript in response to comments provided in the previous version of the manuscript. However, some of the concerns about the language and interpretation still remains in the current form of manuscript.

Major Comments:

1. Since the authors acknowledge in response that 'It is well known that a small fraction (likely less than 10% in most cases) of serum antibodies target linear peptide epitopes' that will be mapped by this 56-mer ArboScan approach. It would be important to include this <10% information in the design

as well as discussion to appropriate guide the readers about the relevance of the information contained in the manuscript.

2. Lines 533-535 and 611-614: The statements are misleading and should be revised. Correlation of binding epitopes and functional PRNT does not mean that these identified binding epitopes are critical neutralizing or functional epitopes. Moreover, 'correlation' does not mean 'association'. Please revise. Suggestion with reference to previous publications on SARS-CoV-2 to few minor S2 epitope, but not with the predominant neutralizing sites in RBD, should not be construed as meaningful relevance. The data in the manuscript and author's responses clearly indicates that the VARscore or the PhIP Arboscan approach is not sensitive to map the functional or neutralizing epitopes that have been primarily defined to be conformational in nature. Please revise and carefully reword to reflect the findings and not overtly interpret some correlations, as that can be misunderstood by the readers.

Reviewer #3 (Remarks to the Author):

Considering my evaluation of the first version and the modifications done in the revised version, I recommend to accept the manuscript for publication.

Please see responses to reviewer comments below in blue text:

REVIEWERS' COMMENTS

Reviewer #1 (Remarks to the Author):

The submission is a revised version where the focus of the manuscript is on the use of an extended peptide phage display library generated from human and zoonotic arboviruses on a T7 phage platform which is named ArboScan. The system was applied against Zika patient materials to identify unique antibody signatures for the specific infection. Additionally, a tailor made algorithm was developed to sieve the peptide data generated to identify key residues associated with the infection post sequencing. The depth of analysis and sample cohorts used is greatly appreciated as it allows for an in depth analysis and comparison.

The authors addressed the comments well.

Reviewer #2 (Remarks to the Author):

Morgenlander et al, phage display approach of 56-mer peptides for Arbovirus family to define the serological fingerprint of antibody reactivity in animals and humans. Furthermore, they correlated some epitope reactivity with functional neutralization activity generated by these viruses. The authors have revised the manuscript in response to comments provided in the previous version of the manuscript. However, some of the concerns about the language and interpretation still remains in the current form of manuscript.

Major Comments:

1. Since the authors acknowledge in response that 'It is well known that a small fraction (likely less than 10% in most cases) of serum antibodies target linear peptide epitopes' that will be mapped by this 56-mer ArboScan approach. It would be important to include this <10% information in the design as well as discussion to appropriately guide the readers about the relevance of the information contained in the manuscript.

The results section includes the sentence: "Both ArboScan and VirScan quantify solely the fraction of anti-viral antibodies that bind the displayed 56-mer peptides, likely a minority of all anti-viral antibodies." We have revised this to add the phrase "likely a minority of all anti-viral antibodies."

In response to prior comments, we have revised the discussion to already include the following limitations: "Finally, PhIP-Seq with 56-mer peptide libraries principally detects the fraction of antibodies that bind to linear epitopes or epitopes completely contained within a displayed peptide, a small fraction of an antibody response induced by viral exposure. Our approach will not detect antibodies to highly conformational/discontinuous epitopes or epitopes containing post-translational modifications. ArboScan therefore does not identify all of the antibody specificities targeted in response to an arbovirus infection." We believe the limitation of only mapping linear peptide epitopes is already specified with this text and do not believe that further text addition to the discussion will add additional value.

2. Lines 533-535 and 611-614: The statements are misleading and should be revised. Correlation of binding epitopes and functional PRNT does not mean that these identified binding epitopes are critical neutralizing or functional epitopes. Moreover, 'correlation' does not mean 'association'. Please revise. Suggestion with reference to previous publications on SARS-CoV-2 to few minor S2 epitope, but not with the predominant neutralizing sites in RBD, should not be construed as meaningful relevance. The data in the manuscript and author's responses clearly indicates that the VARscore or the PhIP ArboScan approach is not sensitive to map the functional or neutralizing epitopes that have been primarily defined to be conformational in nature. Please revise and carefully reword to reflect the findings and not overtly interpret some correlations, as that can be misunderstood by the readers.

We recognize the importance of not overstating the significance of reporting epitope reactivities that correlate with neutralization. Based on prior comments, we have already softened the language in this section with the goal of not misleading the reader to overinterpret these findings. We understand that the terms "correlation" and "association" are not synonymous, but would consider correlation to be a specific type of association. If the intended comment from the reviewer was supposed to be "correlation" does not mean "causation," we believe that text referenced by the reviewer presents these findings as hypothesis generating and does not make a claim of causation as quoted below:

533-535: Unbiased investigation using ArboScan highlighted flavivirus epitopes that associate with neutralization and may rapidly identify candidate epitopes with potential functional importance from less well-studied arboviruses.

611-614: Nevertheless, the identification of a minority of antibody reactivities to nearly every arbovirus may serve as a useful tool to epidemiologists seeking to broadly surveil for arbovirus exposure in diverse global populations, to vaccinologists interested in selecting potential epitope candidates for future study, and to virologists generating hypotheses about antibody responses across related viral species.

Reviewer #3 (Remarks to the Author):

Considering my evaluation of the first version and the modifications done in the revised version, I recommend to accept the manuscript for publication.